# An Upper-Probability-Based Softmax Ensemble Model for Multi-Sensor Bearing Fault Diagnosis

**DOI:** 10.3390/s25226887

**Published:** 2025-11-11

**Authors:** Hangyeol Jo, Yubin Yoo, Miao Dai, Sang-Woo Ban

**Affiliations:** 1Department of Information & Communication Engineering, Graduate School, Dongguk University, Gyeongju 38066, Republic of Korea; johangyeol@dongguk.ac.kr (H.J.); daimiao@dongguk.ac.kr (M.D.); 2Electronics and Telecommunications Research Institute (ETRI), Gwangju 61012, Republic of Korea; yooyubin@etri.re.kr; 3Department of Electronics, Information & Communication Engineering, Dongguk University, Gyeongju 38066, Republic of Korea

**Keywords:** acoustic sensors, AdaBoost, bearing fault diagnosis, convolutional neural networks (CNN), data fusion, ensemble learning, softmax probability, vibration sensors

## Abstract

In bearing fault diagnosis for rotating machinery, multi-sensor data—such as acoustic and vibration signals—are increasingly leveraged to enhance diagnostic performance. However, existing methods often rely on complex network architectures and incur high computational costs, limiting their applicability in real-time industrial environments. To address these challenges, this study proposes a lightweight and efficient multi-sensor ensemble framework that achieves high diagnostic accuracy while minimizing computational overhead. The proposed method transforms vibration and acoustic signals into spectrograms, which are independently processed by modality-specific lightweight convolutional neural networks (CNNs). The softmax outputs from each classifier are integrated using an AdaBoost-based ensemble strategy that emphasizes high-confidence predictions and adapts to sensor-specific misclassification patterns. Experimental results on benchmark datasets—UORED-VAFCLS, KAIST, and an in-house bearing dataset—demonstrate an average classification accuracy exceeding 99.90%, with notable robustness against false positives and missed detections. Furthermore, the framework significantly reduces resource consumption in terms of FLOPs, inference latency, and model size compared to existing state-of-the-art multi-sensor fusion approaches. Overall, this work presents a practical and deployable solution for real-time bearing fault diagnosis, balancing classification performance with computational efficiency without resorting to complex feature fusion mechanisms.

## 1. Introduction

Vibration signals are among the most widely used sensory modalities in condition monitoring and fault diagnosis of mechanical systems such as wind turbines, industrial rotating machinery, and transportation systems. Due to their high responsiveness to dynamic state changes and suitability for real-time monitoring, vibration signals are especially effective in detecting incipient faults [1,2,3]. With the advancement of signal processing and deep learning techniques, vibration-based fault diagnosis methods—particularly those employing convolutional neural networks (CNNs) with attention mechanisms—have demonstrated superior performance in identifying faults in rotating systems [4,5,6,7,8,9,10].

However, in practical industrial environments, sensor signals are frequently distorted due to factors such as differences in sensor mounting positions, structural complexity, and transmission paths [11,12]. Moreover, although bearing vibration is directly caused by faults, it is also influenced by various factors, including elastic deformation of the shaft or supporting structure, assembly errors, and unbalanced loads [13]. For these reasons, various studies have recently been conducted to simultaneously ensure interpretability and adaptive learning capability. Chen et al. [14] proposed a wavelet-based Kolmogorov–Arnold CNN–LSTM model that achieved robustness and interpretability even under noisy environments; however, its structural complexity limited real-time applicability. To alleviate such structural complexity, Zhang et al. [15] improved generalization performance through a spectrum distribution preservation CNN (SDP-CNN), yet the model was restricted to a single-sensor structure, thereby failing to exploit the complementarity among multiple signals. Li et al. [16] analyzed the vibration behavior of gas microturbines under different fuel conditions, demonstrating physical validity, but a data-driven intelligent fusion framework was not incorporated. Wang et al. [17] achieved high accuracy under varying load conditions using an adaptive attention CNN–LSTM model; nevertheless, computational complexity and real-time applicability remained constrained.

Due to the limitation of a single sensor, recent studies have focused on multi-sensor fault diagnosis methods that combine vibration and acoustic signals, as well as techniques such as domain adaptation, transfer learning, and ensemble learning [18,19,20,21]. Acoustic signals are particularly effective in capturing high-frequency components generated by bearing defects and can support early fault detection. However, they are also susceptible to external noise and structural reflections, which may degrade diagnostic accuracy in real-world environments [5].

To enhance diagnostic performance, fusion-based approaches have been proposed, typically classified into data-level, feature-level, and decision-level fusion [19,22,23,24]. Data-level fusion directly integrates raw signals collected from each sensor to extract features, which minimizes information loss but suffers from high computational cost. Feature-level fusion combines independently extracted feature vectors from individual sensors to enhance classification performance; however, the structural complexity increases due to the high-dimensional feature concatenation [24]. In contrast, decision-level fusion improves diagnostic reliability by combining the prediction results of sensor-specific classifiers [24]. Although this approach enables robust prediction without complex feature aggregation, it typically employs static weighted averaging, which fails to reflect sensor-specific reliability differences and misclassification tendencies. Representative studies for each fusion level, along with sensor configurations, key characteristics, and limitations, are summarized in Table 1.

Despite the progress in multi-sensor fusion, several challenges hinder its practical industrial application. First, feature fusion approaches are sensitive to noise or outliers from individual sensors, which may degrade overall model performance. Second, these models often assume equal importance across sensors, disregarding modality-specific reliability. Third, tightly coupled model architectures lack modularity, requiring retraining when sensors are added or replaced. Fourth, diagnostic errors are difficult to trace to either sensor input quality or model limitations, complicating fault mitigation. These factors collectively compromise diagnostic consistency and maintainability.

To address these challenges, this study proposes a novel decision-level ensemble framework that utilizes softmax prediction distributions from sensor-specific CNN classifiers. Instead of integrating raw or feature-level data, the proposed approach uses the probabilistic outputs as candidate representations, allowing for meaningful signal information to be retained—even for misclassified samples. By employing AdaBoost, which iteratively reweights training samples based on classification error, the model enhances its generalization capability and robustness to sensor-specific misclassification tendencies [31,32].

The major contributions of this study are summarized as follows:Uncertainty-Aware Predictive Augmentation: Rather than relying solely on the most probable class, the proposed approach incorporates the top two high-confidence predictions from each sensor-specific model. This mechanism facilitates more stable and reliable decisions, particularly under noisy or ambiguous input conditions.Sensor Reliability-Aware Decision Weighting: The influence of each sensor’s prediction in the final ensemble decision is adaptively adjusted based on its individual classification performance, enabling the system to place greater trust in more reliable sensors.Adaptive Boosting-Based Integration: Through an iterative AdaBoost-based process, the ensemble framework progressively corrects sensor-specific misclassifications, achieving enhanced diagnostic robustness without relying on complex feature extraction or fusion pipelines.Architectural Simplicity and Computational Efficiency: By operating directly on the prediction-level outputs of individual models, the framework eliminates the need for deep or fused intermediate representations. This design significantly reduces computational overhead while maintaining high diagnostic accuracy, rendering the approach suitable for real-time industrial applications.

Comprehensive experimental evaluations conducted on three datasets demonstrate that the proposed framework achieves comparable classification accuracies exceeding 99.9%. Moreover, it consistently outperforms existing state-of-the-art multi-sensor fusion approaches in terms of computational efficiency and scalability.

## 2. Methods

This section describes the overall architecture and procedural framework of the proposed model, which extracts the confident class probabilities from the softmax output distributions of vibration and acoustic signals and uses them as quantitative features for an AdaBoost ensemble classifier. Instead of relying solely on the highest-probability prediction, the model emphasizes high-confidence predictions across multiple class candidates, thereby retaining informative cues even in uncertain or ambiguous cases. The overall framework consists of three modular components, as illustrated in Figure 1.

The first component, the Pre-processing module, applies the Short-Time Fourier Transform (STFT) to the raw vibration and acoustic signals acquired from each sensor, converting them into two-dimensional spectrograms with time-frequency resolution. This transformation is intended to effectively represent the nonlinear and nonstationary fault characteristics typically observed in rotating machinery systems.

The second component, the Feature Extraction module, employs independently constructed CNN classifiers $f^{\left(l\right)}(x^{\left(l\right)})$ for each sensor to output softmax probability distributions $P^{l}$. However, the distribution of softmax outputs is highly dependent on model architecture, network capacity, and dataset characteristics, making a universal fixed *k* impractical. Consequently, it is essential to determine which portion of the softmax distribution provides the most reliable diagnostic information under given conditions. Our analysis indicates that relying solely on the Top-1 prediction may discard valuable alternatives and yield unstable decisions, whereas incorporating additional high-confidence predictions compensates for misclassifications, reduces predictive uncertainty, and enhances inference stability. Simultaneously, indiscriminate expansion of *k* increases input dimensionality and memory requirements while offering only marginal gains in accuracy. To address these trade-offs, the proposed framework implements an adaptive *k*-selection strategy, wherein the number of retained predictions is dynamically determined based on the reliability of the softmax distribution.

Across the three datasets evaluated in this study, the majority of diagnostic improvements were consistently achieved by considering the Top-2 softmax outputs, with negligible additional gains observed beyond this threshold. Accordingly, although the framework is capable of adaptively accommodating varying conditions, the Top-2 configuration is adopted as a practical and empirically validated instantiation of the adaptive strategy, ensuring both diagnostic robustness and computational efficiency.

Each softmax vector is scaled by a reliability weight $\beta_{l}$, which reflects the classification performance of the corresponding sensor model. The most confident class probabilities from each scaled softmax distribution are selected and concatenated to form the final ensemble input vector z. The third component, the AdaBoost-based Decision module, iteratively trains weak classifiers $T_{m}$ using the input vector z, while updating the sample weights $w^{\left(m\right)}$ in each iteration. The final decision classifier C determines the fault class based on the weighted predictions. By directly leveraging softmax distributions, the proposed model compensates for misclassification tendencies and enables reliability-aware decision-making that considers the heterogeneity of different sensor modalities.

Each CNN classifier is independently trained for its respective sensor modality, and final diagnosis is performed in the decision-making stage using the softmax probability distributions from individual classifiers as inputs. Conventional fusion approaches typically require simultaneous and reliable acquisition of signals from all sensors to construct a unified feature vector; consequently, sensor failures or delays directly degrade diagnostic performance due to this structural dependency. In contrast, the proposed framework is capable of decision-making even when the softmax output from a particular sensor is unavailable, relying solely on the outputs of remaining sensors. Importantly, the softmax-based input preserves the probabilistic information of each sensor independently, allowing the utilization of learned feature representations from a single CNN classifier when only one sensor is operational. This design renders the proposed model applicable to both multi-sensor and single-sensor environments.

Vibration and acoustic signals, as representative nonlinear time-series data, reflect the operational state of rotating machinery and exhibit complex, nonlinear variations over time. Even under identical fault conditions, signal amplitude and distribution can vary across sensors due to factors such as attachment position, sensitivity, and structural transmission paths in real industrial settings. These sensor-specific discrepancies may introduce scale mismatches prior to feature extraction, potentially resulting in information imbalance and biased predictions during the learning process.

To address these challenges, Z-score normalization was applied to each sensor signal, standardizing all input data to have a mean of zero and a standard deviation of one. This normalization mitigates discrepancies in sensor representation. Z-score normalization is defined in Equation (1), where each sensor data point x is transformed based on the overall mean μ and standard deviation σ.(1)$$X=\frac{x-\mu }{\sigma }$$

Moreover, bearing fault signals exhibit non-stationary behavior, with their frequency components changing irregularly over time. As a result, relying solely on frequency-domain analysis is often insufficient to capture the complete information inherent in such signals. To facilitate analysis in both the time and frequency domains, the STFT was applied. STFT divides the entire signal into fixed-size windows and performs the Fourier Transform on each segment, enabling the extraction of time-varying spectral features. This approach effectively captures the nonlinear and non-stationary fault patterns present in the signal. STFT can be defined mathematically as shown in Equation (2) [6,33].(2)$$X\left(n,\omega \right)=\sum_{m=0}^{N-1}x\left[n+m\right]s(m)e^{-j\omega m}$$

In this formulation, x[n+m] denotes the input signal, $s\left(m\right)\;$ represents the window function and ω  denotes the frequency. The output of the STFT is represented as a two-dimensional spectrogram containing both time and frequency information.

### 2.1. Architecture of Individual CNN Models

To verify the validity of the proposed method, a CNN model based on a single sensor was implemented. The normalized vibration and acoustic signals were each converted into spectrograms and fed into CNN classifiers of identical structure. As shown in Figure 2, each model consists of three convolutional layers and two fully connected layers. In this study, a CNN architecture commonly used in diagnostic applications was adopted [6,34,35,36]. This architecture follows a general framework that is not restricted to a specific dataset or sensor type and can be extended to various multivariate signals or multiple sensor inputs.

Each sensor-specific CNN classifier consists of three convolutional layers, sequentially applying 128 and 64 filters of size 3 × 3, followed by 32 filters of size 1 × 1. The Rectified linear unit (ReLU) activation function is used in all convolutional layers. To mitigate overfitting and improve generalization performance, a dropout layer was applied after each convolutional layer. The fully connected layers consisted of two hidden layers with 256 nodes each, and a final softmax output layer. The Adam optimizer was employed for model training.

The CNN model used in this study was implemented in a manner consistent with the general single-sensor-based CNN architectures commonly adopted in the literature. No additional structural optimizations or advanced design techniques were applied. This approach is designed to ensure a reliable analysis of the softmax output distributions by applying an identical CNN architecture to each sensor, thereby maintaining consistency when extracting and analyzing the top-ranked probability information.

Although the CNN model itself is based on conventional designs used in existing diagnostic methods, the unique contribution of this study lies in selectively extracting the top-ranked softmax probabilities and integrating them into the AdaBoost ensemble. This approach was designed to systematically address the prediction uncertainties and potential misclassification tendencies that may arise in single-sensor-based classifiers.

### 2.2. Softmax-Based Ensemble Modeling

Vibration and acoustic signals used in rotating machinery diagnostics exhibit distinct sensitivities and frequency response characteristics. Even under identical fault conditions, their time-frequency representations can vary significantly across sensors. These heterogeneous characteristics offer complementary diagnostic information that can mitigate the limitations of single-sensor approaches. However, conventional fusion methods—such as naive aggregation or uniform weighting—fail to fully capture the physical distinctions and reliability variations across sensors, thereby imposing structural limitations on diagnostic performance.

To enable effective multi-sensor fault diagnosis, a computational framework is required that can quantitatively adjust the contribution of each sensor based on its predictive reliability and the uncertainty inherent in its softmax probability distribution. To meet this requirement, this paper proposes an AdaBoost-based ensemble architecture that leverages the upper-ranked softmax probabilities from individual classifiers [37,38]. By integrating softmax outputs and classifier-level accuracy into a weighted ensemble input representation, the proposed model aims to enhance diagnostic accuracy.

To address the variability and complementary characteristics of vibration and acoustic signals, this study adopts an AdaBoost ensemble learning framework, integrating sensor-specific classification outputs based on upper-probability softmax distributions.

(1)Sensor-Wise softmax prediction

The training set is denoted as ${\left\{(x_{i},y_{i})\right\}}_{i=1}^{n}$, where $x_{i}\in R^{d}$ is the spectrogram input of the i-th sample and $y_{i}\in \left\{0,1,2,\ldots ,K\right\}$ is the ground-truth label for $x_{i}$. Here, L denotes the number of sensors used—such as vibration and acoustic—and $f^{\left(l\right)}$ refers to the softmax-based CNN classifier associated with sensor l. Each classifier outputs a class-probability distribution, which is represented in Equation (3):(3)$$P_{i}^{\left(l\right)}=f^{\left(l\right)}\left(x_{i}\right)=\left[p_{i,1}^{\left(l\right)},p_{i,2}^{\left(l\right)}...,p_{i,K}^{\left(l\right)}\right]$$
where $p_{i,K}^{\left(l\right)}$ denotes the predicted probability that input ***x***_i_ belongs to class K, as estimated by classifier $f^{\left(l\right)}$.

(2)Classifier weighting based on accuracy

Let $A_{l}\;$ denote the standalone accuracy of classifier $f^{\left(l\right)}$. Then the normalized ensemble weight $\beta_{l}$ for classifier $f^{\left(l\right)}$ is computed according to Equation (4).(4)$$\beta_{l}=\frac{A_{l}}{\sum_{j=1}^{L}A_{j}}$$

(3)Concatenated ensemble inputs

The weighted softmax outputs from all classifiers are concatenated to form the ensemble input vector, as shown in Equation (5):(5)$$z_{i}=[\beta_{1}P_{i}^{\left(1\right)}∥\beta_{2}P_{i}^{\left(2\right)}∥\ldots ∥\beta_{L}P_{i}^{\left(L\right)}]$$
where ∥ denotes vector concatenation and L represents the number of sensors. This vector $z_{i}$ is then fed into an AdaBoost-based meta-classifier.

(4)AdaBoost training process

The ensemble consists of M weak learners ${\left\{T_{m}\right\}}_{m=1}^{M}$. Each learner is iteratively trained with updated sample weights $w_{i}^{\left(m\right)}.$ The initial sample weight $w_{i}^{\left(1\right)}=\prod_{l=1}^{L}\beta_{l}$ is determined by incorporating the softmax prediction accuracy of each individual classifier. The misclassification rate of the m-th weak classifier, $\varepsilon^{\left(m\right)},$ is computed by Equation (6):(6)$$\varepsilon^{\left(m\right)}=\sum_{i=1}^{n}w_{i}^{\left(m\right)}\cdot I(T_{m}(z_{i})\neq y_{i})$$
where I is an indicator function that returns 1 if the condition is true, and 0 otherwise.

The classifier confidence weight $\alpha_{m}$, which determines the contribution of the m-th classifier, is computed according to Equation (7):(7)$$\alpha_{m}=\frac{1}{2}ln\left(\frac{1-\varepsilon^{\left(m\right)}}{\varepsilon^{\left(m\right)}}\right)$$

The classifier confidence weight $\alpha_{m}$ is used for final voting, as shown in Equation (9), as well as for a sample weight, as shown in Equation (8).

Finally, the sample weight is updated for the next iteration according to Equation (8):(8)$$w_{i}^{\left(m+1\right)}=w_{i}^{\left(m\right)}\cdot exp(\alpha^{\left(m\right)}\cdot I\left(T_{m}\left(z_{i}\right)\neq y_{i}\right))$$

This formulation increases the weight of misclassified samples, allowing the next learner to focus more on difficult cases.

(5)Final Ensemble prediction

The final class prediction is obtained by aggregating the votes of all weak classifiers, weighted by their respective confidences, as shown in Equation (9):(9)$$C\left(x_{i}\right)=\underset{C\in \{1,2,\ldots ,K\}}{\mathrm{argmax}}\left[\sum_{m=1}^{M}\alpha_{m}\cdot I\left(T_{m}\left(z_{i}\right)=c\right)\right]$$

This weighted majority voting scheme ensures that stronger classifiers have a greater influence on the final prediction.

The proposed ensemble model is implemented using the AdaBoost classifier from the scikit-learn 1.3 library in Python 3.8, with 100 estimators, a learning rate of 1.0, and the SAMME.R algorithm, which supports softmax-based boosting. This configuration enables effective ensemble learning with high classification accuracy and computational efficiency.

## 3. Experiments

To comprehensively assess the performance of the proposed AdaBoost-based ensemble model with upper-probability-based softmax outputs, three experimental case studies were conducted using distinct bearing fault datasets. All experiments employed both vibration and acoustic signals as input modalities and were performed under identical hardware conditions (Intel Core i9-10900F CPU (Intel Corporation, Santa Clara, CA, USA), 2.80 GHz, 32 GB RAM).

The first experiment utilized the publicly available University of Ottawa constant load and speed rolling-element bearing vibration and acoustic fault signature datasets (UORED-VAFCLS) [39], a widely adopted benchmark comprising bearing fault data collected from vibration and acoustic sensors. The second experiment involved a multi-sensor rotating machinery fault diagnosis dataset provided by Korea Advanced Institute of Science and Technology (KAIST) [40], collected under ISO-standard test conditions. The third case study was based on data acquired from a custom-built testbed specifically developed for this research.

Each experiment incorporated vibration and acoustic signals acquired under varying conditions and quantitatively evaluated the performance of both the individual CNN classifiers and the proposed ensemble model using multiple performance metrics.

### 3.1. Metrics

Model performance was assessed using four classification metrics—Accuracy, Precision, Recall, and F1-score—defined in Equations (10)–(13). These metrics quantify the alignment between predicted labels and ground truth, with F1-score offering a balanced measure particularly suitable for imbalanced data scenarios. In the Equations, TP, TN, FP, and FN refer to True Positive, True Negative, False Positive, and False Negative, respectively.(10)$$Accuracy=\frac{TP+TN}{TP+TN+FP+FN}$$(11)$$Precision=\frac{TP}{TP+FP}$$(12)$$Recall=\frac{TP}{TP+FN}$$(13)$$F1\text{-}\mathrm{score}=2\times \frac{Precision\times Recall}{Precision+Recall}$$

In addition, real-time applicability and computational efficiency were measured using the number of floating-point operations (FLOPs), inference time, and model size. The total FLOPs were calculated by multiplying the FLOPs of a single CNN by the number of sensor channels due to parallel processing. Inference time represents the latency per sample, while model size indicates the memory footprint of the trained model. These indicators are essential for evaluating the model’s deployment feasibility in embedded or industrial systems.

### 3.2. Dataset Description

To evaluate the generalization performance of the proposed model, three bearing fault datasets with different sensor configurations and operating conditions were utilized. These datasets were collected under varying operational environments and sensor setups. Figure 3 illustrates the testbed employed for data acquisition in the experiments [39,40].

The UORED-VAFCLS dataset provided by the University of Ottawa contains simultaneously collected accelerometer and microphone signals. Both vibration and acoustic signals were recorded at a sampling rate of 42 kHz, and the testbed used for data acquisition is shown in Figure 3a. The dataset is available at https://data.mendeley.com/datasets/y2px5tg92h (accessed on 30 May 2025). Case 2 correspIonds to the rotating machinery bearing fault dataset collected at KAIST, in which accelerometer, acoustic microphone, temperature, and current data were simultaneously acquired. Vibration, temperature, and current were measured at 25.6 kHz, while acoustic signals were recorded at 51.2 kHz. The testbed used for this dataset is presented in Figure 3b and was designed to satisfy the ISO 10816-1 [41] and ISO 8528-10 [42] standard conditions. The dataset is available at https://data.mendeley.com/datasets/ztmf3m7h5x/6 (accessed on 30 May 2025). Case 3 data were collected using a laboratory-built bearing fault diagnosis testbed, as shown in Figure 3c. The system was mounted on a vibration isolation table and consisted of an AC motor, coupling, shaft, bearing housing, and two bearings. The rotational speed was maintained at 1500 RPM, and KBC 6204 open-type bearings were used. During data acquisition, potential factors that could affect the vibration characteristics of rotating machinery—such as shaft misalignment and unbalanced load—were incorporated to simulate realistic industrial fault scenarios. The dataset includes a normal state and six fault types: inner race fault, outer race fault, cage fault, metallic contamination, lubrication deficiency, and shaft misalignment. A vibration sensor (HS 13A231) and an acoustic emission (AE) sensor (PK3I) were employed for data collection, and the bearing specifications and sensor conditions are summarized in Table 2 and Table 3.

The three datasets contain multi-sensor-based bearing fault data collected under different conditions, through which the consistency and generalization performance of the proposed model were comprehensively evaluated.

### 3.3. Pre-Processing

Fault signals generated in bearings exhibit non-stationary characteristics, with vibration properties that vary over time. The frequency components of such signals are distributed differently across time segments, making it difficult to fully capture fault characteristics using information from a single domain. In this study, the STFT was applied to analyze time-varying frequency information. Since STFT extracts local frequency components of a signal using a fixed window [34], it also enables the unification of input formats between vibration and acoustic sensors. Consequently, multi-sensor data with different sampling frequencies can be normalized to a consistent two-dimensional input size, ensuring structural uniformity and learning efficiency of the model. Figure 4 illustrates the process of converting vibration and acoustic signals into time–frequency spectrograms.

Time-domain signals collected from each sensor were converted into spectrograms using the STFT. The window size was set to 0.0343 s, corresponding to approximately 1440 samples at the sampling frequency of 42 kHz. This duration represents one rotational period of the bearing, based on the motor speed of 1750 RPM in the UORED-VAFCLS dataset. Each STFT frame was designed to include vibration and acoustic events occurring within a single mechanical cycle. To maintain consistent temporal resolution, signals from the KAIST and in-house datasets were also segmented using the same 0.0343 s interval for input data construction.

The dataset used in the experiments consisted of 600 samples per class, of which 300 were used for training and the remaining 300 for validation and testing. To ensure statistical reliability, experiments were repeated ten times with different random sample configurations, and the results were reported as the average over all repetitions.

### 3.4. Softmax Output Analysis and Evaluation

The proposed model was evaluated using Top-1 and Top-2 classification accuracies according to the sensor types of the three datasets, in order to verify whether the upper-probability information from the softmax distribution could enhance diagnostic reliability. Table 4 summarizes the mean, minimum, and maximum accuracies based on the Top-1 and Top-2 criteria for the three bearing fault datasets.

Experimental results show that the Top-2 accuracy consistently improved compared to the Top-1 accuracy across all datasets. When relying solely on Top-1 predictions, misclassification occurred in some uncertain samples; however, considering the Top-2 probabilities increased the likelihood that the true label was included among the top-ranked candidates. In particular, for the acoustic sensor in Case 2, the Top-2 accuracy converged to 100%, indicating that the Top-2 probability information carried a high degree of diagnostic validity. A similar trend was observed in the Case 3 dataset, where the Top-2 accuracy remained above 99% in all experiments.

Therefore, in this study, the upper-probability information from the softmax distribution was utilized as ensemble input. Experimental results showed that most misclassifications were corrected when considering up to Top-2 probabilities, while extending beyond Top-3 resulted in less than a 0.05% improvement in accuracy. This indicates that the most informative diagnostic cues are concentrated within the top two probability values, and the Top-2 input effectively captures the uncertainty between the primary candidate classes. Although the optimal value of *k* may vary depending on the characteristics of the dataset, the experiments in this study demonstrated that the Top-2 configuration provides an effective trade-off between diagnostic performance and computational efficiency.

Table 5 presents the effectiveness and generalization performance of the proposed model compared with individual sensor-based diagnostic models across the three datasets. Each value is reported as the mean and standard deviation over ten repeated experiments.

The experimental results show that the proposed ensemble model achieved higher mean accuracy than the single-sensor models for all datasets, while exhibiting significantly lower standard deviations, indicating more stable predictive performance. In particular, for the Case 2 dataset, the ensemble model achieved a mean accuracy of 99.98%, demonstrating an almost saturated level of diagnostic precision.

This performance improvement can be attributed to the utilization of upper-probability information from the sensor-specific softmax distributions and the compensation of sensor-specific misclassification tendencies through AdaBoost-based learning. In other words, the uncertainty inherent in individual sensor models was effectively mitigated by the probability-based ensemble, thereby enhancing both diagnostic stability and generalization performance.

### 3.5. Comparative Performance Results

To evaluate the generalization performance and real-time applicability of the proposed model, a performance comparison was conducted with existing fusion-based diagnostic methods, namely MFF-GBDT, CDTFAFN, and MSFF-Net. MFF-GBDT applies an ensemble strategy at the decision level of multi-sensor inputs, employing an approach similar to that of the proposed model. CDTFAFN represents a feature-level fusion structure that achieved high accuracy using the Ottawa dataset, while MSFF-Net is a lightweight model that utilizes one-dimensional spectral inputs of acoustic and vibration signals.

Table 6 presents the comparison results for the Case 1, Case 2, and Case 3 datasets. The proposed model demonstrated significantly higher accuracy and lower standard deviation compared with the existing methods. In particular, it achieved an accuracy improvement of approximately 18.8 percentage points over MFF-GBDT, with a substantial reduction in prediction variance.

These results indicate that the softmax upper-probability-based decision-level ensemble effectively compensates for sensor-specific misclassification tendencies while achieving higher reliability with a simpler structure compared to feature-level fusion methods.

Furthermore, to assess the real-time applicability of the proposed model, its computational efficiency and embedded deployment feasibility were evaluated using the Case1–Case3 datasets. Table 7 compares the FLOPs, inference time, and memory requirements of each fusion model. The proposed model achieved an inference time of 0.00002 s, 0.0003 MFLOPs, and a memory usage of 0.068 MB, demonstrating a substantially lighter architecture than CDTFAFN (34.578 MFLOPs, 200 MB) and MSFF-Net (236.497 MFLOPs, 27.5 MB). Notably, both CDTFAFN and MSFF-Net exceed the 0.51 MB SRAM capacity, potentially leading to high computational overhead in MCU-based environments. In contrast, the proposed model can operate within the memory constraints of a Raspberry Pi Pico 2 (dual-core Cortex-M33, 150 MHz × 2, Raspberry Pi Ltd., Cambridge, UK) and exhibits low computational load, confirming its feasibility for real-time processing.

These results demonstrate that, while the proposed model can be embedded in the same manner as other fusion models, it exhibits superior characteristics in terms of system resource efficiency and real-time inference stability.

### 3.6. Missing-Data Experiment

In practical industrial bearing diagnosis, temporary data loss from certain sensors may occur due to hardware connection faults, transmission errors, or sampling omissions. Such missing inputs can destabilize sensor reliability and consequently degrade the diagnostic confidence of both single-sensor and fusion-based models.

The proposed softmax ensemble structure, which utilizes upper-probability information, processes the probability distributions of each sensor independently and performs weighted combination through AdaBoost-based learning. This design enables compensation using the remaining sensor predictions even when some sensor data are missing.

To verify this structural robustness, missing-data experiments were conducted by intentionally removing partial segments from vibration and acoustic sensor inputs. The missing rates were set to 10%, 20%, and 30%, and the removed segments were zero-padded to induce signal discontinuity. The missing data ratios were determined with reference to previous studies [43,44].

Table 8 presents the comparison results between single-sensor models with missing inputs (Acou-Miss: single-sensor model with Acoustic signal missing, Vib-Miss: single-sensor model with Vibration signal missing) and the proposed ensemble model (Ens) across the three datasets. The proposed model consistently maintained significantly higher accuracy than the single-sensor counterparts under all missing data conditions.

The proposed model maintained diagnostic accuracy above 98% across all datasets, even with a missing rate of 30%. In contrast, the performance of single-sensor models dropped by more than 20 percentage points as the missing rate increased. These results indicate that the proposed softmax ensemble structure based on upper-probability information effectively preserves diagnostic stability through complementary fusion, even under sensor-missing conditions. Therefore, the proposed framework demonstrates robust performance and reliability in environments subject to sensor faults or data loss.

## 4. Conclusions

In this study, an AdaBoost-based ensemble diagnostic framework utilizing upper-probability information from the softmax distribution was proposed. Conventional multi-sensor fusion models require complex feature combination processes and high computational costs to integrate sensor data, thereby limiting their applicability in real-time environments. The proposed model performs reliability-based fusion at the decision level by leveraging the softmax probability outputs of individual sensors, achieving both high accuracy and computational efficiency. In particular, it minimizes structural complexity and computational load while maintaining diagnostic precision, reducing FLOPs, inference time, and model size by more than 90%, thereby experimentally demonstrating its feasibility for real-time industrial applications. These results indicate that the proposed softmax-based ensemble approach serves as a practical alternative that achieves high reliability with a simpler architecture compared to existing fusion methods.

Future work will focus on mitigating performance degradation in samples with low softmax confidence by applying prediction reliability calibration techniques and exploring dynamic relationship modeling to capture inter-sensor dependencies. Furthermore, the framework will be extended to support diverse sensor configurations and asynchronous inputs, aiming to enhance the generality of lightweight AI systems for real-time industrial fault diagnosis.

## Figures and Tables

**Figure 1 sensors-25-06887-f001:**
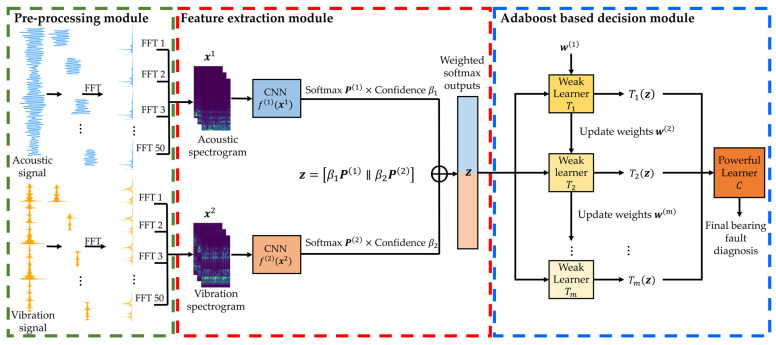
Architecture of the proposed AdaBoost ensemble model using upper-probability-based softmax outputs.

**Figure 2 sensors-25-06887-f002:**
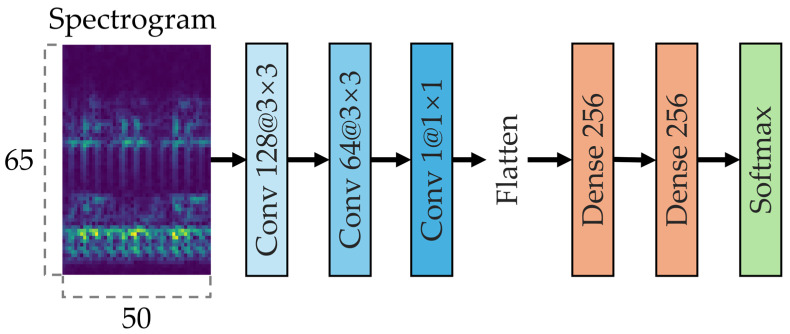
CNN model structure for single-sensor diagnosis.

**Figure 3 sensors-25-06887-f003:**
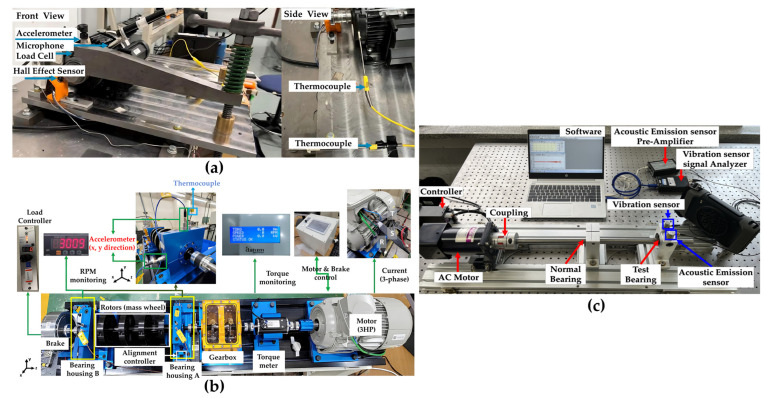
Experimental testbeds: (**a**) UORED-VAFCLS [39], (**b**) KAIST [40], (**c**) In-house.

**Figure 4 sensors-25-06887-f004:**
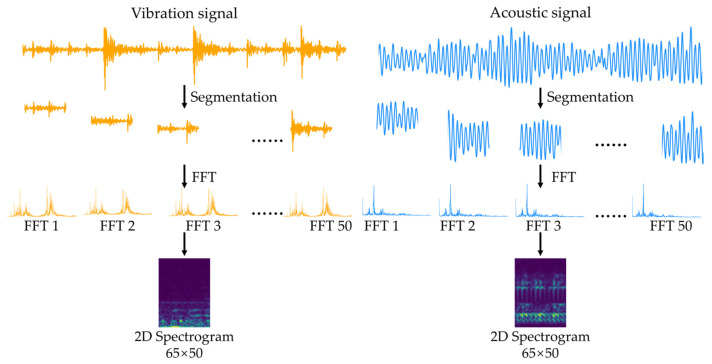
The process of generating spectrograms from vibration and acoustic sensor signals.

**Table 1 sensors-25-06887-t001:** Summary of major studies and characteristics according to fusion levels.

Approach	Inputs		Main Contribution
	Number of Sensors	Sensor Type	
Data fusion (Wang, X et al. [19])	2	Accelerometer/Microphone	The diagnostic performance was enhanced through sensor data fusion based on 1D-CNN, resulting in an increase in computational load due to signal-level integration.
Data fusion (Wang, J et al. [25])	3	Acceleration	By integrating raw signals from multiple locations using a 2D-CNN, information loss was reduced, while the computational cost remained high.
Data fusion (Tao, J et al. [26])	3	Magnet acceleration	Feature representations from multiple sensors were fused using a deep belief network (DBN), leading to improved diagnostic accuracy accompanied by increased structural complexity.
Data fusion (Wang, S et al. [27])	2	Acceleration	Multi-sensor signals decomposed by variational mode decomposition (VMD) were analyzed using an ultra-lightweight GoogLeNet (UL-GoogLeNet), achieving high accuracy but not reflecting sensor-specific reliability.
Feature fusion (Song, R et al. [12])	2	Accelerometer	Spatiotemporal fusion with entropy-based weighting enhanced representational capability, while multiple stages of feature extraction resulted in an increased number of preprocessing steps.
Feature fusion (Yan, X et al. [18])	2	Accelerometer/Microphone	Spatial, temporal, and frequency features were fused using dual-scale attention, resulting in improved feature integration but increased complexity due to high-dimensional combination and intermediate fusion stages.
Feature fusion (Dai, M et al. [21])	2	Accelerometer/Microphone	Two frequency-domain signals were fused via FFT and diagnosed using a lightweight 1D-CNN, achieving high accuracy and low computational cost, though the simple fusion method may cause information redundancy.
Decision fusion (Shao, H et al. [28])	7	Vibration	Diagnostic accuracy was improved through predictive fusion based on a stacked wavelet auto-encoder, while the weight configuration remained empirical.
Decision fusion (Liu, Z et al. [29])	2≤ Sensors	Air pressure	Multidimensional features from multiple sensors were fused and used in ensemble learning to diagnose braking system faults, resulting in effective fault identification but increased model complexity and computational cost.
Decision fusion (Xu, X et al. [30])	4	Vibration/Acoustic	Dynamic decision-level fusion adaptively calibrated multi-signal classification results using statistical features from a variational autoencoder, improving diagnostic accuracy and reliability but increasing real-time computational load and model complexity.

**Table 2 sensors-25-06887-t002:** Specification of testing bearing on the in-house dataset.

Bearing Type	Pitch Diameter	Ball Diameter	Number of Balls
KBC 6204	9.52 mm	36 mm	7

**Table 3 sensors-25-06887-t003:** Sampling and measurement characteristics of vibration and AE sensors.

	Vib	AE
Sampling frequency	12,000 Hz	7168 Hz
Frequency range	0.5~10,000 Hz	15,000~40,000 Hz
Resonant Frequency	-	28,000 Hz

**Table 4 sensors-25-06887-t004:** Comparison of diagnostic accuracy according to sensor type and decision level.

Dataset	Sensor Type	Decision Level	Mean Accuracy (%)	Min. Accuracy (%)	Max. Accuracy (%)
Case 1	Acoustic	Top-1	99.54	99.10	99.80
		Top-2	99.92	99.80	100
	Vibration	Top-1	99.52	99.20	99.80
		Top-2	99.94	99.90	100
Case 2	Acoustic	Top-1	99.93	99.88	99.97
		Top-2	100	100	100
	Vibration	Top-1	99.98	99.96	99.99
		Top-2	100	100	100
Case 3	Acoustic emission	Top-1	99.80	98.68	99.85
		Top-2	100	100	100
	Vibration	Top-1	99.94	99.90	99.97
		Top-2	100	100	100

**Table 5 sensors-25-06887-t005:** Classification accuracy comparison between single-sensor models and the proposed ensemble model across three datasets.

Dataset	Acoustic	Vibration	Proposed Model
Case1	99.52 ± 0.0022%	99.52 ± 0.0019%	99.93 ± 0.0003%
Case2	99.93 ± 0.0003%	99.98 ± 0.0002%	99.99 ± 0.0001%
Case3	99.80 ± 0.0006%	99.94 ± 0.0002%	99.95 ± 0.0003%

**Table 6 sensors-25-06887-t006:** Comparative classification performance of the proposed model and existing fusion methods across three bearing fault datasets.

Dataset	Method	Accuracy (%)	Precision (%)	Recall (%)	F1-Score (%)
Case 1	MFF-GBDT	81.75 ± 0.0420	82.05 ± 0.0420	81.04 ± 0.0420	81.17 ± 0.0420
	CDTFAFN	99.90 ± 0.0060	99.90 ± 0.0060	99.90 ± 0.0060	99.90 ± 0.0060
	MSFF-Net	99.91 ± 0.0003	99.91 ± 0.0003	99.91 ± 0.0003	99.91 ± 0.0003
	Proposed model	99.90 ± 0.0004	99.93 ± 0.0004	99.93 ± 0.0004	99.93 ± 0.0004
Case 2	MFF-GBDT	98.38 ± 0.0035	98.41 ± 0.0036	93.36 ± 0.0035	98.38 ± 0.0038
	CDTFAFN	99.95 ± 0.0008	99.95 ± 0.0007	99.95 ± 0.0008	99.95 ± 0.0008
	MSFF-Net	99.25 ± 0.0188	99.40 ± 0.0142	99.25 ± 0.0187	99.23 ± 0.0192
	Proposed model	99.98 ± 0.0004	99.99 ± 0.0003	99.99 ± 0.0003	99.99 ± 0.0003
Case 3	MFF-GBDT	78.00 ± 0.0710	79.15 ± 0.0710	78.00 ± 0.0710	87.20 ± 0.0710
	CDTFAFN	99.99 ± 0.0006	99.99 ± 0.0006	99.99 ± 0.0006	99.99 ± 0.0006
	MSFF-Net	99.91 ± 0.0002	99.91 ± 0.0002	99.9 ± 0.0002	99.91 ± 0.0030
	Proposed model	99.99 ± 0.0003	99.99 ± 0.0003	99.99 ± 0.0003	99.99 ± 0.0004

**Table 7 sensors-25-06887-t007:** Comparison of computational complexity and resource efficiency among multi-sensory fusion models.

Multi-Sensory Fusion Methods	FLOPs (M)	Model Size (MB)	Inference Time (s)		
			Case 1	Case 2	Case 3
MFF-GBDT	0.0015	0.757	0.00006	0.00004	0.00006
CDTFAFN	34.5781	200.000	0.01680	0.01670	0.01670
MSFF-Net	236.4972	27.500	0.00175	0.00200	0.00190
Proposed model	0.0003	0.068	0.00002	0.00002	0.00003

**Table 8 sensors-25-06887-t008:** Diagnostic accuracy (%) under different missing sensor rates.

Method	Missing Rate (%)											
	10		20		30		10		20		30	
	Acou-Miss	Ens	Acou-Miss	Ens	Acou-Miss	Ens	Vib-Miss	Ens	Vib-Miss	Ens	Vib-Miss	Ens
Case1	90.29	98.26	82.47	98.98	74.29	98.52	88.15	96.20	80.77	98.20	74.41	98.18
Case2	91.76	99.64	83.36	99.75	75.72	99.95	90.71	99.69	82.49	99.85	75.93	99.85
Case3	91.60	97.14	82.99	98.57	74.37	99.98	91.53	95.71	82.69	98.57	74.14	99.67

## Data Availability

Data will be made available on request.

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
