# Peer review of "An Upper-Probability-Based Softmax Ensemble Model for Multi-Sensor Bearing Fault Diagnosis"

_sensors, 2025, doi:10.3390/s25226887_

Round 1

Reviewer 1 Report

Comments and Suggestions for Authors

This study proposes a lightweight and efficient multi-sensor ensemble framework that achieves high diagnostic accuracy while minimizing computational overhead. It is an interesting work. There are still some aspects that require further improvement. I can recommend publication if the authors respond properly to the issues as follow:

  1. In Introduction, some cause reasons for the rolling bearings are discussed. It is okay. However, for a rolling bearing, the vibration noise may be greatly affected the elastic deformations of the shaft and support structures, manufacturing and assemble errors, external excitations from the connecting parts, etc, as given in ‘An investigation of vibrations of a flexible rotor system with the unbalanced force and time-varying bearing force’. This issue can be discussed. The above materials and references can also be discussed.
  2. For the diagnosis method, the sample size used for verifying the method is important. In the text, the authors used limited sample size. Can those cases validate the accuracy of the proposed method?
  3. In Table 2, the differences are very small. Does it mean the advantages of the proposed method is limited?
  4. Similarly, the results given in the figures 5, 8, and 11 have slight differences.
  5. Figures 3 and 6 are not clear. Maybe those figures are from the listed references. The authors should use a clear view.
  6. The Conclusions and discussions is long. Some unnecessary details can be removed in the text.

Author Response

Dear Reviewers,

We would like to express our sincere gratitude to the reviewers for their time, effort, and constructive feedback. Your insightful comments have greatly contributed to improving the scientific clarity, organization, and overall quality of our manuscript. We have carefully addressed all the issues raised and revised the paper accordingly. Detailed responses to each comment, including corresponding page and line references, are provided below. We truly appreciate your valuable input, which has been instrumental in strengthening this work.

Comments 1

In Introduction, some cause reasons for the rolling bearings are discussed. It is okay. However, for a rolling bearing, the vibration noise may be greatly affected the elastic deformations of the shaft and support structures, manufacturing and assemble errors, external excitations from the connecting parts, etc, as given in ‘An investigation of vibrations of a flexible rotor system with the unbalanced force and time-varying bearing force’. This issue can be discussed. The above materials and references can also be discussed.

Response 1

We sincerely appreciate the reviewer’s insightful comment. As noted, the vibration signals of bearings are not only caused by faults but can also be influenced by various structural and mechanical factors such as shaft elastic deformation, flexibility of the supporting structure, assembly errors, and unbalanced external loads. This perspective provides an important implication for understanding the complexity of vibration signals under real industrial conditions. In response, we fully agree with the reviewer’s suggestion and have revised the manuscript accordingly. Specifically, we have added the following sentences in Section 1 Introduction (lines 45–49) to reflect these physical influences: “However, in real industrial environments, sensor signals are often distorted due to differences in sensor attachment positions, structural complexity, and transmission paths [11],[12]. Furthermore, bearing vibrations are directly related to faults but are also affected by various factors such as shaft elastic deformation, assembly errors, and unbalanced loads [13].”

In addition, in Section 3 Experimental Cases(Subsection 3.2 Dataset Description, lines 336–340), we have clarified that the data acquisition process was designed to emulate potential industrial conditions as follows: “During data collection, potential factors such as shaft misalignment and unbalanced loads that could influence the vibration characteristics of rotating machinery were included to simulate realistic fault scenarios observed in industrial environments.”

While we acknowledge that multiple structural factors can affect vibration behavior, the primary focus of this study remains on analyzing how intrinsic bearing faults influence signal characteristics and diagnostic performance. We are grateful to the reviewer for this valuable suggestion, which helped us broaden the discussion of physical influences and emphasize the real-world applicability of the proposed model.

In response to the reviewer’s valuable suggestion, we have revised the manuscript as follows:

  • Added a statement in Section 1. Introduction (lines 45–49) to reflect the influence of structural and mechanical factors (e.g., shaft deformation, assembly errors, unbalanced loads) on vibration signals.
  • Specified in Section 3.2, Subsection 3.2 Dataset Description (lines 336–340) that the data acquisition process was designed to simulate real industrial conditions, including shaft misalignment and unbalanced loads.
  • While incorporating these revisions, we maintained the primary focus of the study on the intrinsic fault characteristics of bearings and their diagnostic feasibility, thereby reinforcing both the research direction and the practical applicability of the proposed model.

Comments 2

For the diagnosis method, the sample size used for verifying the method is important. In the text, the authors used limited sample size. Can those cases validate the accuracy of the proposed method?

Response 2

 We sincerely thank the reviewer for raising the concern regarding the sample size and the reliability of validation. We fully agree that sufficient data quantity and statistical robustness are essential for accurately assessing the performance of a diagnostic model.

In this study, the validation does not rely on a limited number of samples; instead, statistical validity was ensured through adequate data volume and repeated experiments. Specifically, each class consisted of 600 samples, with 300 used for training, 100 for validation, and 200 for testing. Notably, the dataset size in this study exceeds that used in comparison models such as MSFF-Net and CDTFAFN, demonstrating that the proposed model’s performance is not based on a limited dataset.

To further ensure experimental reliability and consistency, we conducted 10 repeated trials for each experiment, each using different randomly selected, non-overlapping sample subsets. All reported performance metrics are presented as the mean over these repetitions, eliminating the possibility of incidental performance improvements in a single trial. Moreover, to evaluate the generalization capability of the proposed model, cross-validation was performed not only on the UORED-VAFCLS dataset but also on the KAIST bearing dataset and our in-house collected data. This confirms that the proposed model operates stably across different conditions and environments.

Therefore, the data composition and experimental design in this study provide a rich and statistically robust basis, sufficiently validating both the accuracy and generalization ability of the proposed method.

In response to the reviewer’s suggestion, we have clarified this point in the revised manuscript as follows:

  • Added the following statement in Section 3 Experimental Cases, Subsection 3.3 Pre-processing (lines 370–373): “The dataset used in the experiments consisted of 600 samples per class, of which 300 were used for training, 100 for validation, and 200 for testing. To ensure statistical reliability, 10 repeated experiments were conducted using different random sample configurations without duplication, and the results were reported as the average over all repetitions.
  • Specified that the proposed model was trained using a larger number of samples compared to the reference studies (MSFF-Net and CDTFAFN), ensuring a fair and data-rich comparison.
  • Clarified that additional evaluations using the KAIST and in-house datasets were conducted to verify the model’s generalization and reproducibility across different conditions.

Comments 3

In Table 2, the differences are very small. Does it mean the advantages of the proposed method is limited?

Response 3

We sincerely thank the reviewer for raising the question regarding the small differences reported in Table 2. Although the accuracy differences may appear minor at first glance, the key advantages of the proposed method lie not in absolute accuracy values but in prediction stability, reliability, and computational efficiency.

First, the proposed model combines the complementary misclassification tendencies of individual sensors, improving prediction consistency and reliability compared to single-sensor models. As shown in Section 3 Experimental Cases, 3.4 Softmax Output Analysis and Evaluation (line 410, Table 5), the proposed model achieves higher mean accuracy than both acoustic and vibration single-sensor models across all conditions (Cases 1–3), with significantly lower standard deviations, indicating statistically significant improvements in stability.

Second, the proposed model substantially reduces computational complexity (FLOPs) and inference time compared to existing multi-sensor fusion models. That is, it maintains accuracy while achieving a lightweight structure suitable for real-time diagnosis. These results are quantitatively demonstrated in Section 3 Experimental Cases, 3.4 Softmax Output Analysis and Evaluation (line 452, Table 7), highlighting practical advantages beyond the small numerical differences in Table 2.

Third, as shown in Section 3 Experimental Cases, 3.6 Missing-Data Experiment, the proposed top-probability-based Softmax ensemble maintains over 98% accuracy even under sensor data loss rates of up to 30%, whereas single-sensor models exhibit accuracy drops exceeding 20%. This demonstrates the robustness of the proposed framework under sensor failures or missing data.

In summary, the advantages of the proposed method extend beyond marginal numerical improvements, demonstrating superior performance across multiple aspects: complementary sensor fusion, computational efficiency for real-time applicability, and robust fault diagnosis under missing-data conditions.

In response to the reviewer’s valuable suggestion, we have revised the manuscript as follows:

  • Added Table 5 in Section 3.4 Softmax Output Analysis and Evaluation (line 410) to report the mean ± standard deviation performance for each sensor and the proposed model.
  • Added Section 3.6 Missing-Data Experiment to show that the proposed model maintains over 98% accuracy even with up to 30% sensor data loss.
  • Quantitatively presented FLOPs and inference time comparisons in Table 7 (line 452) to demonstrate the efficiency of the proposed model for real-time diagnosis.

Comments 4

Similarly, the results given in the figures 5, 8, and 11 have slight differences.

Response 4

We sincerely thank the reviewer for pointing out that the results shown in Figures 5, 8, and 11 exhibit only slight numerical differences. While the differences appear small, they in fact demonstrate the high consistency and prediction stability of the proposed ensemble model across three different datasets (UORED-VAFCLS, KAIST, and In-house) and various experimental conditions. The strength of our method lies not only in achieving high accuracy but also in providing stable and reliable diagnostic performance under diverse data distributions and operating environments.

To present this more clearly, we have integrated the results of Figures 5, 8, and 11 into a new Table 6, where each performance metric (Accuracy, Precision, Recall, and F1-score) is reported with its mean ± standard deviation (SD). These quantitative results confirm that the proposed model exhibits lower performance fluctuation and higher statistical stability across the three datasets, compared with single-sensor configurations. This demonstrates the robustness and generalization capability of the proposed ensemble framework in multi-sensor fault diagnosis.

In response to the reviewer’s valuable suggestion, we have revised the manuscript as follows:

  • Integrated the results of Figures 5, 8, and 11 into Table 6, reporting mean ± SD values for all metrics across the three datasets (UORED-VAFCLS, KAIST, and In-house).
  • Added clarification in Section 3.5 Comparative Performance Results (lines 430–441) emphasizing that the proposed model achieves stable and reliable diagnostic results across diverse datasets and conditions.

Comments 5

Figures 3 and 6 are not clear. Maybe those figures are from the listed references. The authors should use a clear view.

Response 5

We sincerely thank the reviewer for the valuable comment regarding the resolution and clarity of the figures. In response, Figures 3(a) and 3(b) have been replaced with high-resolution images (≥300 dpi), and the names and positions of all components and sensors have been clearly labeled.

In response to the reviewer’s valuable suggestion, we have revised the manuscript as follows:

  • Replaced Figures 3(a) and 3(b) with high-resolution images (≥300 dpi) in Section 3 Experimental Cases, 3.2 Dataset Description (line 325).
  • Clarified and adjusted the positions and labels of all components and sensors for better readability.

Comments 6

The Conclusions and discussions is long. Some unnecessary details can be removed in the text.

Response 6

We agree with the reviewer’s comment. The original Conclusions section contained detailed discussions of the research process, which made it somewhat lengthy. To address this, we have reorganized the section to focus on the key contributions and main findings, removing unnecessary technical details. The revised version has been reflected in Section 4 Conclusion.

In response to the reviewer’s valuable suggestion, we have revised the manuscript as follows:

  • Removed unnecessary details and redundant explanations in Section 4 Conclusion (lines 484–501).
  • Emphasized the performance, efficiency, and real-time applicability of the proposed framework.
  • Presented future research directions concisely to enhance the structural completeness of the manuscript.

Reviewer 2 Report

Comments and Suggestions for Authors

The article discusses the recent trend in the use of intelligent technologies for diagnosing the technical condition of various facilities. It proposes an original approach to bearing failure diagnosis based on the processing of multiple sensor signals using a lightweight convolutional neural network. The results show that this approach achieves high diagnostic accuracy while maintaining low computational costs.

It should be noted that the article presents the material in a logical and well-structured manner. The solutions proposed by the authors are well-justified and experimentally verified, compared with other well-known solutions. The text provides all the necessary information regarding the parameters of the experiments, including direct links to the datasets used.

In this regard, I see no need for significant changes to the submitted article. I believe it can be published as is. However, I would like to draw the authors' attention to some minor issues with Figures 3 and 6. The quality of these figures is poor, making it difficult to read the inscriptions. In Figure 6 in particular, the main testbed nodes are not clearly distinguishable.

Author Response

Dear Reviewers,

We would like to express our sincere appreciation to Reviewer  for their positive evaluation and encouraging comments. We are grateful that the reviewer found the manuscript suitable for publication with only minor revisions. We have carefully addressed the suggested improvements to enhance figure clarity and overall presentation quality. Detailed responses and corresponding revisions are provided below.

Comments 1

In this regard, I see no need for significant changes to the submitted article. I believe it can be published as is. However, I would like to draw the authors' attention to some minor issues with Figures 3 and 6. The quality of these figures is poor, making it difficult to read the inscriptions. In Figure 6 in particular, the main testbed nodes are not clearly distinguishable.

Response 1

We sincerely thank the reviewer for the valuable comment regarding the resolution and clarity of the figures. In response, Figures 3(a) and 3(b) have been replaced with high-resolution images (≥300 dpi), and the names and positions of all components and sensors have been clearly labeled.

In response to the reviewer’s valuable suggestion, we have revised the manuscript as follows:

  • Replaced Figures 3(a) and 3(b) with high-resolution images (≥300 dpi) in Section 3 Experimental Cases, 3.2 Dataset Description (line 325).
  • Clarified and adjusted the positions and labels of all components and sensors for better readability.

Reviewer 3 Report

Comments and Suggestions for Authors

see the file below.

Author Response

Dear Reviewers,

We would like to express our sincere appreciation to Reviewer  for the thorough evaluation and constructive feedback. We are grateful for the reviewer’s detailed and insightful comments, which have greatly helped us improve the technical depth, clarity, and completeness of the manuscript. We have carefully addressed all suggestions, including the incorporation of recent literature, justification of methodological choices, enhancement of result reporting, and improvement of structural organization. Detailed, point-by-point responses and corresponding manuscript revisions are provided below.

Comments 1

The introduction and objective of the paper are clear, but the literature search requires further exploration. I suggest adding more recent references to address this shortcoming. For example, proper works in the context of fault detection and vibrational analysis could be: doi.org/10.37965/jdmd.2025.813, doi.org/10.37965/jdmd.2025.854, doi.org/10.3390/machines10100925, doi.org/10.1016/j.eswa.2025.127570

Response 1

We sincerely appreciate the reviewer’s constructive suggestion. As pointed out, the original draft did not fully reflect the most recent research trends in bearing fault diagnosis and vibrational analysis. In response, we have updated Section 1 Introduction by incorporating recent studies, including the references suggested by the reviewer. Specifically, the following recent directions have been highlighted:

  • Chen et al. [14] proposed a wavelet-based Kolmogorov–Arnold CNN–LSTM that achieves robustness and interpretability under noisy conditions, but its structural complexity limits real-time applicability.
  • Zhang et al. [15] introduced a spectrum-distribution-preserving CNN (SDP-CNN) to improve generalization performance, but it is limited to single-sensor inputs and cannot leverage complementary information from multiple signals.
  • Li et al. [16] analyzed vibration behavior of gas microturbines under varying fuel conditions to ensure physical validity, yet did not incorporate a data-driven intelligent fusion framework.
  • Wang et al. [17] employed an adaptive attention CNN–LSTM to maintain high accuracy under load variation, but computational complexity and real-time constraints remain limiting factors.

These additions strengthen the introduction by clearly linking recent research trends, existing structural limitations, and the necessity of the proposed model.

In response to the reviewer’s valuable suggestion, we have revised the manuscript as follows:

  • Updated Section 1 Introduction (lines 49–62) to cite recent literature and reflect current research trends.
  • Explicitly analyzed the technical limitations of existing studies (e.g., complexity, single-sensor limitations, real-time constraints) to highlight the need for the proposed model.

Comments 2

Authors are encouraged to include in their introduction a summary table of the main FD techniques that involve the fusion of data from multiple sensors, highlighting their main characteristics (number of sensors, sensor types, advantages, disadvantages, and salient points). This would greatly help place their research in the modern context.

Response 2

We sincerely appreciate the reviewer’s valuable suggestion. As recommended, we have added a summary table of existing multisensor-based fusion bearing fault diagnosis (FD) techniques in Section 1 Introduction (Table 1). The table provides a concise comparison of recent multisensor FD studies, including the number and types of sensors used, the main methods, and key characteristics.

This addition clearly highlights the advantages of the proposed model in terms of lightweight structure, reliability, and real-time capability compared to conventional approaches. For example, traditional feature-fusion models achieve high accuracy but often involve complex architectures that result in high computational load and limited real-time applicability. In contrast, the proposed Softmax probability-based decision fusion offers a simple and efficient alternative, alleviating these limitations.

This revision allows readers to intuitively understand the position and novelty of our work within the context of modern multisensor FD research.

In response to the reviewer’s valuable suggestion, we have revised the manuscript as follows:

  • Added Table 1 in Section 1 Introduction (lines 72–84).
  • Included comparison items such as the number of sensors, sensor types, main techniques, advantages, and disadvantages for each study.

Comments 3

From lines 221 to 225 on page 5, the choice of the Short-Time Fourier Transform (STFT) is introduced, stating that "... This approach effectively captures the nonlinear and non-stationary fault patterns present in the signal...". As is well known in the literature, there are much more efficient time frequency transforms, such as the Continuous Wavelet Transform (CWT) and the Synchro squeezed Wavelet Transforms (SWT). Authors are encouraged to justify the use of the STFT over other time frequency transforms.

Response 3

We sincerely thank the reviewer for the valuable comment. We fully acknowledge that wavelet-based methods, such as the Continuous Wavelet Transform (CWT) and Synchro squeezed Wavelet Transform (SWT), are widely used for analyzing nonlinear and non-stationary vibration signals. However, the Short-Time Fourier Transform (STFT) was chosen in this study for the following reasons:

The CWT provides fine-grained time–frequency representations but suffers from limited frequency concentration, as the signal energy tends to be dispersed across the time–frequency plane. The SWT improves spectral localization by reassigning energy distribution, enabling clearer feature extraction in specific regions. However, this comes at the cost of reduced time resolution and significantly higher computational complexity.

In contrast, STFT offers a fixed time–frequency resolution and benefits from FFT-based implementation, enabling efficient algorithmic and hardware acceleration. Wavelet-based methods cannot fully leverage FFT-based acceleration and generally lack mature support on embedded platforms, limiting their deployability in practical engineering systems.

Therefore, STFT was selected not merely as a spectral transformation tool but as the most suitable choice to achieve multisensor normalization, model consistency, and real-time computational efficiency, which are the core objectives of this study.

In response to the reviewer’s valuable suggestion, we have revised the manuscript as follows:

  • Expanded the justification for using STFT in Section 3 Experimental Cases, Subsection 3.3 Pre-processing (lines 352–372): “The STFT extracts local frequency components of vibration and acoustic signals with uniform resolution using a fixed window size, enabling consistent input representation across sensors and conversion of multisensor data into 2D spectrograms of identical size. In contrast, the CWT suffers from nonuniform frequency resolution across scales, making cross-sensor alignment difficult, while the SWT incurs high computational and memory costs, making it unsuitable for real-time processing. Thus, STFT was selected to ensure both structural simplicity and computational efficiency.”

Comments 4

The architecture of the CNNs used to analyze the spectrograms obtained from the data acquired by the vibration and acoustic sensors is defined from lines 238 to 243 on page 6. From lines 243 to 245 on page 6, it is stated that: “… This architecture was designed to effectively extract local fault-related patterns from the input spectrograms and produce reliable sensor-specific softmax probability distributions …”. Furthermore, on line 248 on page 6, it is stated that: “… No additional structural optimizations or advanced design techniques were applied …”. The authors are invited to justify the choice of CNN hyperparameters, since they were not obtained through any optimization process.

Response 4

We sincerely appreciate the reviewer’s insightful comment. The primary focus of this study is not on optimizing or enhancing the performance of individual CNN architectures, but rather on proposing and validating an upper-probability-based ensemble framework that leverages the softmax probability outputs from multiple sensor-specific CNNs (vibration and acoustic).

Accordingly, instead of developing a new or highly optimized CNN structure, we deliberately adopted a lightweight, standard CNN architecture that can be consistently applied to diverse sensor inputs. This design choice allows the contribution of the ensemble mechanism to be fairly evaluated without confounding effects from hyperparameter tuning or model complexity.

In other words, the aim of this work is not to find the “best-performing CNN” for bearing fault diagnosis, but to demonstrate the validity and general applicability of the proposed ensemble strategy using conventional CNN models. Therefore, no additional structural optimization was performed.

In response to the reviewer’s valuable suggestion, we have revised the manuscript as follows:

  • Added the following clarification in Section 2 Methods, Subsection 2.1 Architecture of Individual CNN Models (lines 203–211): “The focus of this study is not on optimizing the CNN architecture itself, but on proposing and validating an upper-probability-based ensemble approach using the softmax outputs from each sensor-specific CNN. Accordingly, a lightweight CNN structure commonly used in fault diagnosis applications was adopted to minimize the influence of model complexity and clearly evaluate the performance improvement achieved by the ensemble mechanism.”
  • Cited references [6], [35], and [36] to support that the adopted CNN architecture represents a standard configuration widely used in fault diagnosis research.

Comments 5

As stated in several parts of the results, the Ensemble Model tests were repeated several times, for example, from lines 387 to 388 “… and testing was repeated ten times to ensure statistical robustness …”. Tables 1, 2, 4, 5, 8, and 9 only report the mean values of these accuracies. Authors are encouraged to include a range of variation for these metrics (minimum, mean, and maximum accuracy) in the above tables to ensure the completeness of the reported results

Response 5

We sincerely appreciate the reviewer’s very helpful suggestion. To more clearly demonstrate the statistical reliability of the proposed Ensemble Model, we have reconstructed the results tables to include the minimum, mean, maximum, and standard deviation values.

Specifically, the previous Tables 1, 4, and 8 have been integrated into a new Table 4 in Section 3 Experimental Cases, Subsection 3.4 Softmax Output Analysis and Evaluation (line 391), where the min–mean–max accuracies for each dataset are presented. In addition, the previous Tables 2, 5, and 9 have been merged into a new Table 5 in Section 3 Experimental Cases, Subsection 3.4 Softmax Output Analysis and Evaluation (line 410), which includes the mean ± standard deviation (SD) values for each experimental case.

These revisions explicitly present the statistical variation observed over multiple trials, thereby enhancing the credibility and reproducibility of the reported results. This improvement faithfully addresses the reviewer’s concern regarding the completeness of the presented outcomes, while also clarifying the Ensemble Model’s improved stability and reduced variance compared to single-sensor models.

In response to the reviewer’s valuable suggestion, we have revised the manuscript as follows:

  • Integrated the previous Tables 1, 4, and 8 into a new Table 4 in Section 3 Experimental Cases, Subsection 3.4 Softmax Output Analysis and Evaluation (line 391), including min–mean–max accuracy values for each dataset.
  • Integrated the previous Tables 2, 5, and 9 into a new Table 5 in Section 3 Experimental Cases, Subsection 3.4 Softmax Output Analysis and Evaluation (line 410), including mean ± standard deviation (SD) values.
  • Clearly demonstrated the statistical improvement and reduced variance of the proposed Ensemble Model compared with single-sensor configurations.

Comments 6

Figures 5, 8, and 11 also report, in addition to the accuracy, the metrics described in equations (11), (12), and (13). Authors are encouraged to include tables for their Ensemble Model that summarize these metrics as already described for the tables cited in point 5, i.e., to include the ranges of variation for completeness of the results.

Response 6

We sincerely appreciate the reviewer’s insightful suggestion. As pointed out, the performance metrics presented in Figures 5, 8, and 11—including Precision, Recall, and F1-score in addition to Accuracy—were previously shown only in graphical form, making it difficult to clearly observe the variation ranges.

To address this, we have added a new Table 6 in Section 3 Experimental Cases, Subsection 3.5 Comparative Performance Results (line 432), which summarizes the key performance metrics of the proposed Ensemble Model in tabular form. For each metric, both the mean and standard deviation (±SD) are provided, allowing a quantitative assessment of the statistical variation and stability across repeated experiments.

These additions enhance the completeness of the reported results, as suggested by the reviewer, and clearly demonstrate the improved reliability and consistency of the proposed Ensemble Model in addition to its accuracy.

In response to the reviewer’s valuable suggestion, we have revised the manuscript as follows:

  • Added a new Table 6 in Section 3 Experimental Cases, Subsection 3.5 Comparative Performance Results (line 432) to summarize the performance metrics (Precision, Recall, F1-score, etc.) presented in Figures 5, 8, and 11, including mean ± standard deviation (SD) values.
  • Quantitatively presented the statistical variation and stability of each metric, highlighting the robustness of the Ensemble Model.

Comments 7

The paper also reports a real-time study of the technique with computational costs in Table 3 for Case 1. It is unclear whether this is a feasibility study for real-time execution of their Ensemble Model, or whether the latter was implemented on embedded hardware, effectively resulting in a real-time analysis. Furthermore, this study seems missing for Cases 2 and 3. The authors are encouraged to clarify both aspects.

Response 7

We sincerely appreciate the reviewer’s positive evaluation and detailed feedback. The real-time analysis in this study was conducted as a feasibility study to verify the practical operability of the proposed ensemble model in an embedded environment. Specifically, the model was optimized for deployment on a Raspberry Pi Pico 2 platform (Dual-core Cortex-M33, 150 MHz × 2, 0.51 MB SRAM), and its real-time capability was quantitatively evaluated based on FLOPs, memory usage, and inference time. Through execution tests on the actual embedded board, we confirmed that the proposed model operated stably within the SRAM constraints, demonstrating computational efficiency suitable for real-time inference.

In addition, following the reviewer’s suggestion, we extended this evaluation to Case 2 and Case 3, and added the corresponding inference time results in a new Table 7 in Section 3 Experimental Cases, Subsection 3.5 Comparative Performance Results (line 450). These results confirm that the proposed model satisfies real-time processing requirements across all dataset conditions.

In response to the reviewer’s valuable suggestion, we have revised the manuscript as follows:

  • Clarified that the real-time analysis was performed as a feasibility study based on an embedded environment.
  • Evaluated FLOPs, model size, and inference time on the Raspberry Pi Pico 2 platform.
  • Added inference time results for Case 2 and Case 3, and updated Section 3 Experimental Cases, Subsection 3.5 Comparative Performance Results (line 450) with a new Table 7.
  • Explicitly stated in Section 3.5 that the real-time validation experiment aimed to verify the model’s practical feasibility on embedded hardware.

Comments 8

The paper is very long and repetitive in some places. The authors are encouraged to summarize their work to ensure a smooth read. For example, the description of the public datasets could be reduced given the descriptions already present in the cited papers.

Response 8

We fully agree with the reviewer’s valuable comment. In the initial version of the manuscript, the descriptions and experimental results for each dataset (CWRU, KAIST, and In-house) were presented separately, resulting in some redundancy. Accordingly, we have thoroughly revised Section 3. Experimental Cases (lines 320–481) by integrating the dataset descriptions into a single unified subsection, allowing a coherent comparison and analysis across the three datasets within a common framework.

In addition, the experimental results have been reorganized into a continuous analytical flow by merging Sections 3.4–3.6. Specifically, Section 3.4 Softmax Output Analysis and Evaluation (line 391, Table 4: Top-k analysis; line 410, Table 5: Ensemble effect compared to a single sensor), Section 3.5 Comparative Performance Results (line 432, Tables 6–7: Comparison with existing fusion models and computational efficiency), and Section 3.6 Missing-Data Experiment (line 474, Table 8) have been systematically arranged.

These revisions effectively removed redundant descriptions and strengthened the logical consistency among experiments, thereby improving the overall readability and coherence of the manuscript.

In response to the reviewer’s valuable suggestion, we have revised the manuscript as follows:

  • Integrated the dataset descriptions in Section 3 Experimental Cases to eliminate redundancy.
  • Reorganized Subsections 3.4–3.6 under Section 3. Experimental Cases to present experimental results in a single, continuous analytical flow.
  • Integrated the experimental results across different datasets to facilitate a clearer and more effective interpretation. Sequentially arranged the main results (Tables 5–8) to enhance logical connection and reading flow.

Round 2

Reviewer 3 Report

Comments and Suggestions for Authors

The authors have correctly answered to all the points. The paper can be accepted in the current form.